# Mapping protein selectivity landscapes using multi-target selective screening and next-generation sequencing of combinatorial libraries

Si Naftaly[1], Itay Cohen[1], Anat Shahar[2], Alexandra Hockla[3], Evette S. Radisky[3] & Niv Papo[1]

Characterizing the binding selectivity landscape of interacting proteins is crucial both for elucidating the underlying mechanisms of their interaction and for developing selective inhibitors. However, current mapping methods are laborious and cannot provide a sufficiently comprehensive description of the landscape. Here, we introduce a novel and efficient strategy for comprehensively mapping the binding landscape of proteins using a combination of experimental multi-target selective library screening and in silico next-generation sequencing analysis. We map the binding landscape of a non-selective trypsin inhibitor, the amyloid protein precursor inhibitor (APPI), to each of the four human serine proteases (kallikrein-6, mesotrypsin, and anionic and cationic trypsins). We then use this map to dissect and improve the affinity and selectivity of APPI variants toward each of the four proteases. Our strategy can be used as a platform for the development of a new generation of target-selective probes and therapeutic agents based on selective protein–protein interactions.

[1] Department of Biotechnology Engineering and the National Institute of Biotechnology in the Negev, Ben-Gurion University of the Negev, Beer-Sheva, Israel. [2] The National Institute for Biotechnology in the Negev (NIBN), Beer-Sheva, Israel. [3] Department of Cancer Biology, Mayo Clinic Comprehensive Cancer Center, Jacksonville, Florida 32224, USA. These authors contributed equally: Si Naftaly, Itay Cohen. Correspondence and requests for materials should be addressed to N.P. (email: papo@bgu.ac.il)

A defining characteristic of protein–protein interactions (PPIs) is the binding selectivity landscape of the interacting proteins[1-3], which relates the amino acid sequence to the affinity of a protein toward its target. Comprehensively mapping this landscape is crucial both for understanding the mechanisms and evolutionary origins of selective PPIs and for protein engineering purposes, e.g., for designing selective binders and/or inhibitors for target proteins[4-7]. The binding selectivity landscape of each protein in a certain PPI is characterized by the interfacial residues of the protein, such that point mutating these residues can help determine the contribution of each residue to target selectivity—or in a protein with a broad selectivity spectrum to the selectivity of the protein to each of its putative targets individually. The binding selectivity landscape usually comprises of four types of key interface residues. Hot-spot residues are a few[8] interface residues that are highly relevant for a specific PPI, i.e., they contribute almost 75% of the total free energy of binding ($\Delta\Delta G_{bind}$) of the protein to its partner[9-11]. Mutating hot-spot residues therefore decreases the affinity of the protein to a specific partner—but not necessarily to others. Cold-spot residues[1,12-14] are interface residues occupied by suboptimal amino acids, such that mutating them increases the binding affinity of the protein to a specific partner. Selectivity-switch residues[15,16] are interface residues in which a point-mutation simultaneously decreases the affinity of the protein to one partner and increases its affinity to another. Finally, correlated-selectivity residues[17,18] are interface residues that work together to increase the selectivity of the protein to one specific partner. Such residues are especially difficult to characterize with conventional methods because only a double mutation (one mutation in each residue) can change the affinity of the protein to a certain partner.

Methods for mapping protein selectivity landscapes typically include mutating candidate residues and testing the resulting changes in affinity[4,19]. Despite considerable advancements in recent years[8,20-22], currently available methods still demonstrate several caveats that hinder our ability to develop, inter alia, selective inhibitors for clinically important proteins. For instance, alanine scanning and similar classical approaches[23-29] can test only a subset of all possible mutants, are time-consuming and laborious, require protein purification and binding affinity measurements for each mutant, and most importantly, focus chiefly on hot-spot residues. Modern approaches can overcome some of these caveats by employing protein library display and sorting technologies, which rapidly explore all possible (hot- and cold-spot) mutations and qualitatively map the contribution of each residue to the affinity of a protein toward its target[30-37]. However, as only several hundred clones of the sorted libraries are ultimately sequenced, these methods do not comprehensively characterize the entire library and, to date, they cannot identify correlated-selectivity residues. A more recent approach employed next-generation sequencing (NGS) to guide protein and synthetic small-molecule optimization[4,7,38-42], effectively improving the binding affinity and selectivity, and generating binding epitopes de novo[6,7,19,40,41,43,44]. However, this and most other currently available approaches generate high-affinity (but not necessarily selective) binders, and, in the few studies designed to generate selective binders[7,40,41,43,44], the methodology was limited to improving discrimination between only two target proteins that have different binding epitopes. Significantly, some broad-spectrum proteins may have many potential targets with binding epitopes that have high-sequence homology and structural similarity.

In the current study, we present a novel, single-step approach for comprehensively mapping the binding selectivity landscape of proteins (including hot-spot, cold-spot, selectivity-switch, and correlated-selectivity residues) using a combination of experimental multi-target selective library screening and in silico NGS analysis. To test our approach in a real-life context, we chose to map the binding selectivity landscape of a broad-spectrum trypsin inhibitor, namely, the human amyloid protein precursor inhibitor (APPI; a member of the human Kunitz-domain family of serine protease inhibitors[45]), to each of the four human serine proteases—kallikrein-6 (KLK6), mesotrypsin, anionic trypsin, and cationic trypsin—all of which share high-sequence homology and structural similarity. Then, we used this landscape to improve both the selectivity and affinity of APPI variants to each protease, which we evaluated through inhibition studies using the purified proteins.

We recently used a yeast-surface display (YSD) platform—a powerful directed evolution protein engineering technology[30,46-50]—to generate APPI-3M[51]: a triple-mutant APPI (M17G/I18F/F34V) whose affinity to mesotrypsin, anionic trypsin, and cationic trypsin is comparable [$K_i = 89.8 \pm 0.23$ pM, $1.47 \pm 0.02$ pM, and $4.96 \pm 0.25$ pM for mesotrypsin, anionic trypsin, and cationic trypsin, respectively[51]], whereas its affinity to KLK6 is lower by three orders of magnitude [$K_i = 1.09 \pm 0.12$ nM[51]]. These features render APPI-3M an optimal model scaffold for engineering binding selectivity; its lack of selectivity toward mesotrypsin, anionic trypsin, and cationic trypsin is a good starting point for manipulating its relative selectivity, while its lower selectivity toward KLK6 makes it a good target for engineering selectivity switches.

Our study design is demonstrated in Supplementary Figure 1. We began by generating a yeast-displayed APPI-3M library including clones with single-residue random mutations in the binding interface (i.e., in the APPI binding loop) and clones with multiple-residue random mutations both within and beyond the binding interface (i.e., in the APPI scaffold and binding loop). Then, we divided the four proteases into combinatorial pairs (six combinations) and sorted the YSD APPI-3M library for variants with differential selectivity toward each protease in each pair. We then used NGS to sequence these fractions and analyzed them computationally. Consequently, the sorted APPI-3M mutant library fractions were rich in affinity- and selectivity-enhancing mutations; of these, we identified the most highly selective APPI mutations based on their ability to inhibit—as soluble proteins—each of the four proteases. To the best of our knowledge, this is the first report of a platform that can provide such a rich PPI binding selectivity landscape.

## Results

**Selecting APPI variants with improved selectivity.** We began by generating a library of APPI-3M clones using both site-directed random mutagenesis of the APPI-3M binding loop (residues 11–18, except invariant Cys-14) and error-prone PCR amplification of the entire coding sequence. This design yielded a 'naive' APPI-3M library of 3.5 million variants, each with 0–2 amino acid mutations. Then, using our YSD system, we expressed each of these variants on the surface of yeast cells and used fluorescence-activated cell sorting (FACS) to quantify their binding to each of the four (soluble) serine proteases (mesotrypsin, KLK6, anionic trypsin, and cationic trypsin). We introduced the yeast-displayed naive library to pairs of serine proteases, each labeled with a different fluorescent dye (Alexa Fluor-650 or Alexa Fluor-488; i.e., a pairwise selective screen, Fig. 1a), at concentrations optimized for each pair to achieve an equivalent distribution of staining intensities (Fig. 1b). The library was sorted to isolate ~1 million variants per sorted fraction (sorting gate), with increased selectivity toward each of the four serine protease targets versus its paired protease. Subsequent FACS analyses showed clear enrichment of the binding

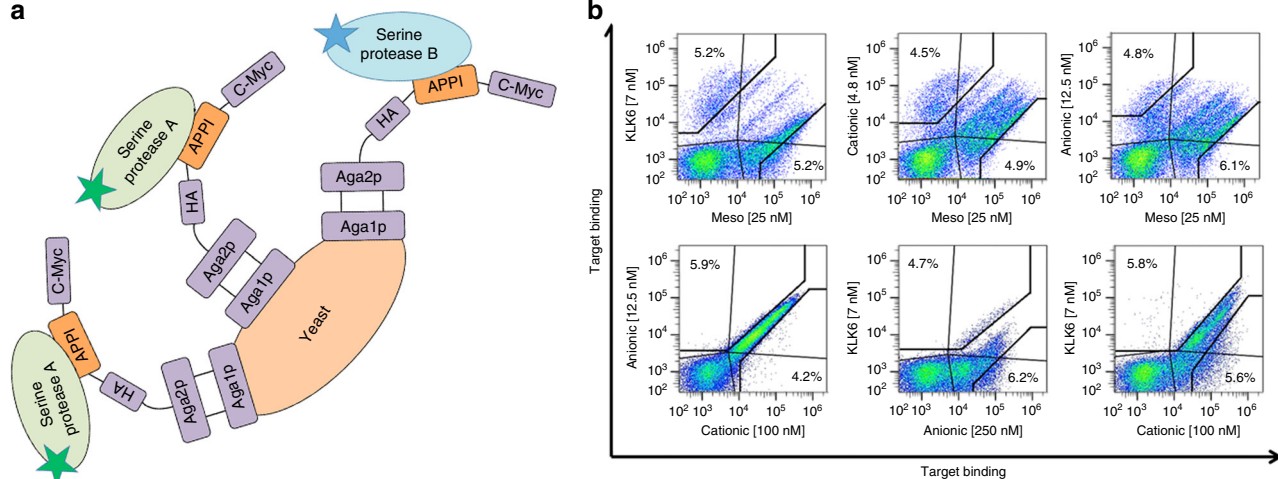

**Fig. 1** Yeast-surface display of APPI-3M. **a** Schematic drawing of the pairwise selective screen using the YSD system. A naive library of mutated APPI-3M variants was displayed on the yeast cell surface and presented to pairs of proteases. Each protease in the pair (denoted A or B) was labeled with a different fluorescent dye—Alexa Fluor-650 or Alexa Fluor-488 (represented by green and blue stars, respectively). **b** Pairwise selective screen. Flow cytometry sorting was used to screen the library to isolate APPI-3M variants, with enhanced selectivity toward each of the four serine proteases (Meso: mesotrypsin; KLK6; Anionic: anionic trypsin; Cationic: cationic trypsin). In each sort, two variant populations were collected inside the black gates, yielding sorted library populations of protease-selective APPI-3M variants. Green and blue colors represent a high and low cell densities, respectively

population for each individual protease (Supplementary Figure 2), confirming selectivity improvements of the sorted library fractions.

**Mapping hot and cold spots and selectivity switches**. To map the binding selectivity landscape of APPI-3M to each of the four serine proteases, we used Illumina Miseq to perform a high-throughput sequencing of APPI-3M gene fragments from the sorted and naive libraries. We then used this sequencing data to identify single and double amino acid substitutions in the APPI-3M sequence that had modulated its selectivity toward each of the four serine proteases. In each sorted fraction, the average number of read pairs per sequenced library was 1 million; of these, 95% of the sequenced read pairs passed quality filtering and integration and were thus translated to amino acid sequences and aligned to the sequence of APPI-3M. Because we were only interested in amino acid substitutions (and not in insertions or deletions), we analyzed only sequences of the same length as that of APPI-3M and determined a threshold value of 100 reads for variants with a single-amino acid substitution and 10 reads for variants with a double-amino acid substitution. To correlate between the abundance of a variant and its target selectivity, we determined the enrichment ratio of each variant, which we defined as the frequency of a certain mutation in the sorted library fraction divided by the frequency of that mutation in the naive library. Thus, we assumed that mutations that increase the selectivity of each variant to its putative target (mesotrypsin, KLK6, anionic trypsin, or cationic trypsin) will be more abundant in the sorted library fraction than in the naive library (enrichment ratio >1), while mutations that decrease selectivity will be less abundant in the sorted library than in the naive library (enrichment ratio <1).

We first characterized the effect of single-amino acid substitution on target selectivity. To this end, we created a heatmap for each sorted library fraction (Fig. 2 and Supplementary Figure 3), using the enrichment ratio as a measure of binding selectivity. Then, we used this map to identify (i) hot spots, defined as APPI-3M residues, in which most mutations decreased the binding selectivity to one target protease versus another, (ii) cold spots, in which most mutations increased the binding selectivity; and (iii) selectivity switches, in which a single mutation decreased the

selectivity to one target protease and increased the selectivity toward another.

Our analysis revealed that residue 15 in APPI-3M is a general hot spot for binding human serine proteases, as all mutations in this residue, except a substitution to Lys, decreased its binding affinity toward all four proteases (Fig. 2 and Supplementary Figure 3). The analysis also revealed two clear cold spots: most mutations in residue 13 increased binding selectivity toward mesotrypsin versus all other proteases (Fig. 2a and Supplementary Figure 3A, dashed line), while most mutations in residue 17 increased binding selectivity toward KLK6 versus all other proteases (Fig. 2b and Supplementary Figure 3B, dashed line). These two selectivity-switch residues (13 and 17) enable a selectivity shift from three proteases toward a single, different protease (either mesotrypsin or KLK6). In addition, most mutations in residue 17 increased the selectivity of APPI-3M toward anionic trypsin and cationic trypsin as compared with mesotrypsin (Fig. 2c, d and Supplementary Figure 3C and D, lower dashed line), while most mutations in residues 11 and 18 increased the selectivity toward anionic trypsin and cationic trypsin as compared with KLK6 (Fig. 2c, d and Supplementary Figure 3C and D, upper dashed line). For example, we found that residues 11 and 17 are selectivity switches for mesotrypsin and KLK6, respectively (Fig. 2, Supplementary Figure 3 and Table 1), as mutating the residue in position 11 (originally Thr) from His to Ile (Fig. 2a, d, white arrows) switched the selectivity from anionic trypsin to mesotrypsin by a factor of $69 \times 10^3$ and mutating the residue in position 17 (originally Gly) from Glu to Arg (Fig. 2b, c, black arrows) switched the selectivity from cationic trypsin to KLK6 by a factor of $7 \times 10^3$.

**Mapping correlated-selectivity residues**. Next, we turned to identify the effects of double-amino acid substitutions in APPI-3M on the selectivity toward each of the four serine proteases. The first steps in this process (quality filtration and integration, translation, alignment, and enrichment ratio calculations) were similar to those described above for single-amino acid analyses. Most double-mutant APPI-3M variants that increased the selectivity toward one serine protease versus all others increased the selectivity toward KLK6 [note that the affinity of the parental

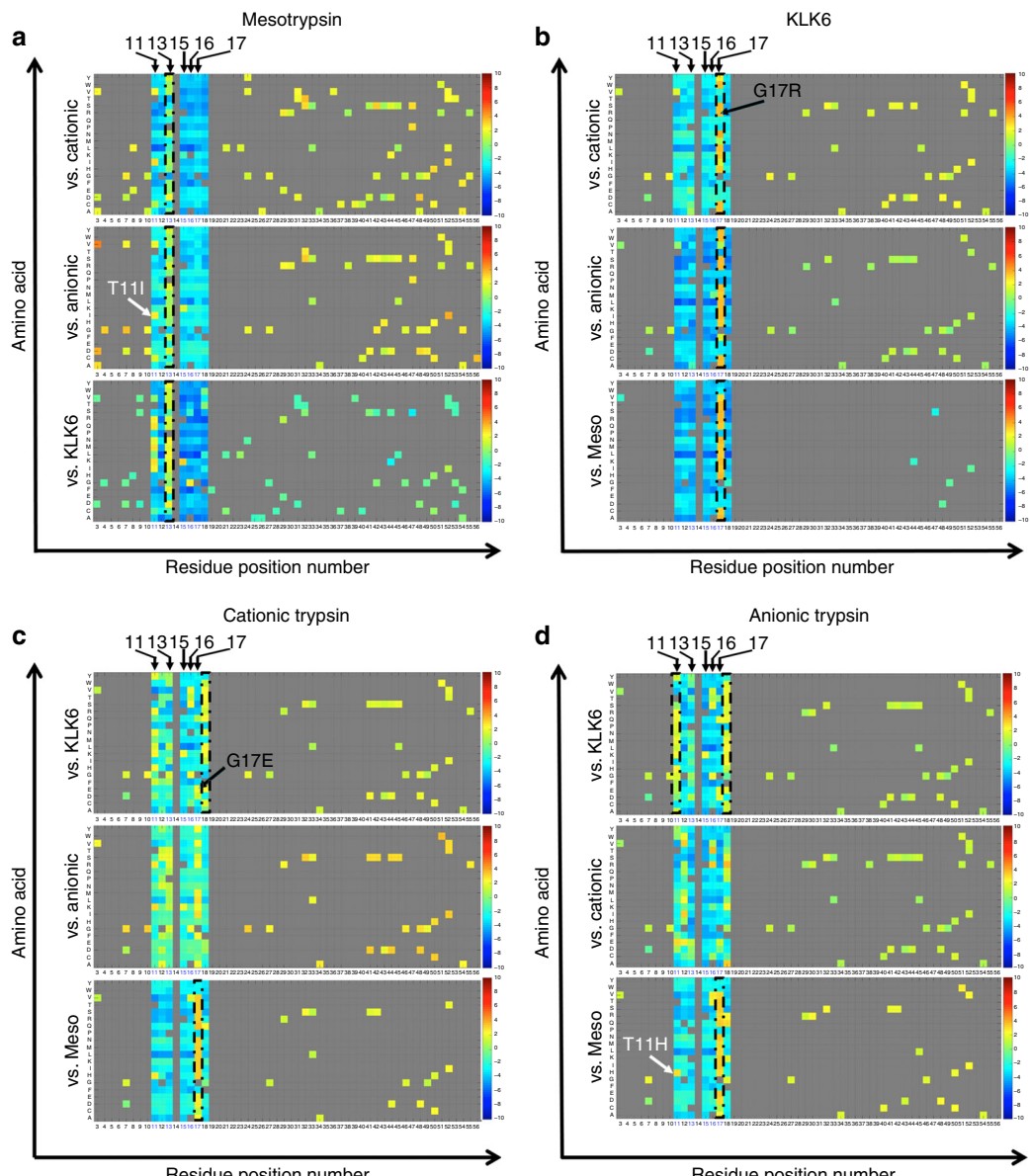

**Fig. 2** Single mutation selectivity landscape of APPI-3M. The colors in the heatmaps indicate the enrichment ratio (defined as the frequency of a certain mutation in the sorted library divided by its frequency in the naive library) and represent the effect of a single-amino acid substitution on APPI-3M selectivity toward one serine protease (**a**: mesotrypsin, **b**: KLK6, **c**: cationic trypsin, and **d**: anionic trypsin) versus the other three. The different colors of the heatmaps correspond to the scale shown on the right of each panel and indicate log2 of the enrichment ratio [yellow and red: positive (increased selectivity); green: negative (decreased selectivity)]. The position of the substituted amino acid is shown on the X axis and the substituting amino acid is shown on the Y axis. Meso: mesotrypsin; Cationic: cationic trypsin; Anionic: anionic trypsin. See Supplementary Figure 3 for further details

APPI-3M to KLK6 was two orders of magnitude lower than to anionic and cationic trypsin and one order of magnitude lower than to mesotrypsin[51]], and these variants had mutations in residues 11 and 17 (Supplementary Table 1). To elucidate the effects of correlated residues and of residues 11 and 17 (Fig. 3c), in particular on the selectivity toward KLK6, we predicted the total effect of each pair of mutated residues (i.e., the effect of all mutations in these two residues; see Methods) and illustrated the results as heatmaps (Fig. 3a and Supplementary Figure 4). Variants in which both residues 11 and 17 were mutated demonstrated an increased selectivity (enrichment ratio >1) toward KLK6 versus the three other proteases. Therefore, we generated additional heatmaps to estimate the effect of specific pair residues (all pair combinations of residues 11 and 17, Fig. 3b). These heatmaps (Fig. 3b) revealed that many combinations of double-amino acid substitutions increased the selectivity toward KLK6

versus the three other proteases, including a combination of either Val, Ala, Pro, or Ser at residue 11 with either Ala, Arg, or Ser at residue 17 (enrichment ratio >1). For instance, the combination of Val at residue 11 and Arg at residue 17 increased the total selectivity of APPI-3M toward KLK6 by a factor of $4 \times 10^9$ (calculated as the multiplication of the three relative selectivities: $\sim59 \times 10^3$-fold versus mesotrypsin, $\sim364$-fold versus cationic trypsin, and $\sim170$-fold versus anionic trypsin; Supplementary Table 2). Similarly, the combination of Ser at residue 11 and Arg at residue 17 increased the total selectivity by a factor of $7 \times 10^7$ ($\sim37 \times 10^3$-fold versus mesotrypsin, $\sim24$-fold versus cationic trypsin, and $\sim85$-fold versus anionic trypsin).

**Validating the selectivity changes using soluble inhibitors**. To validate the results of the NGS computational analysis, we

**Table 1 Selectivity of APPI-3M variants with mutations at selectivity-switch residues toward human serine proteases**

| Mutation | Target A | Target B | Enrichment ratio, target A | Enrichment ratio, target B | Selectivity[a] |
|---|---|---|---|---|---|
| T11H | Mesotrypsin | Anionic trypsin | 0.07 | 7.39 | **1** |
| T11I | | | 5.91 | 0.01 | **$69.22 \times 10^3$** |
| G17E | KLK6 | Cationic trypsin | 0.12 | 4.88 | **1** |
| G17R | | | 12.50 | 0.08 | **$6.65 \times 10^3$** |

[a]Selectivity is defined as the fold change in the enrichment ratio for target A, divided by the fold change in the enrichment ratio for target B

generated and purified the soluble forms of APPI-3M variants, in which the mutation was located at selectivity-switch residues on the APPI loop (Supplementary Figure 5). These variants included the mutations T11I (for which the NGS analysis predicted a selectivity switch from anionic trypsin to mesotrypsin), T11H (predicted switch from mesotrypsin to anionic trypsin), G17R (predicted switch from cationic trypsin to KLK6), and G17E (predicted switch from KLK6 to cationic trypsin) (Table 1). Then, we evaluated the affinity of these four purified APPI-3M variants to each serine protease by measuring the degree to which they inhibit the ability of each protease to hydrolyze its substrate [benzyloxycarbonyl–Gly–Pro–Arg–p–nitroanilide (Z-GPR-pNA) for mesotrypsin, anionic trypsin, cationic trypsin, and BOC–FSR–MSC for KLK6]. We determined the inhibition constant ($K_i$) of each of these interactions by quantifying the slow tight binding behavior (Supplementary Figure 6). The experimental results indeed correlated well with those of the NGS analysis (Table 2).

**Positions 11 and 17 in the APPI-3M sequence are correlated**. As residues in positions 11 and 17 of the APPI-3M sequence increased the selectivity of APPI-3M toward KLK6, we elucidated the interactions between different amino acids at these positions by generating and purifying representative single- and double-mutant APPI-3M variants. We chose the KLK6-selective T11V/G17R and T11S/G17R double-mutant variants (see Fig. 3), and their corresponding single-mutant selectivity-switch variants T11V, T11S, and G17R (see Tables 1 and 2). We tested the affinity of the soluble forms of these five variants to each of the four serine proteases in a competitive inhibition assay (Table 3) and, based on the extracted $K_i$ values, we determined the selectivity of each variant toward KLK6 and compared it with the selectivity of the unmodified APPI-3M (Table 4).

The amino acid substitution that most increased the total selectivity of APPI-3M toward KLK6 was T11V/G17R, followed by G17R and finally, T11S/G17R. The individual substitutions T11V and T11S did not improve the selectivity toward KLK6, rather they somewhat decreased it (Table 4). These results suggest that residues 11 and 17 are correlated-selectivity residues, which act together to increase target selectivity. To further test this hypothesis, we conducted a double-mutant cycle analysis[52], in which we used the selectivity values of KLK6 with the two double-mutant variants and their single variants (T11V/G17R, T11S/G17R, T11V, T11S, and G17R, Table 4) to calculate the selectivity strength between two mutated residues (i.e., the coupling energy, $\Delta\Delta G_{int}$; Supplementary Figure 7). Indeed, in both double mutations, the $\Delta\Delta G_{int}$ values were non-zero, indicating that residues 11 and 17 interact with each other to cooperatively affect the selectivity toward KLK6.

To gain insight into the structural basis of the observed selectivity changes, we attempted to crystallize the APPI-3M-T11V/G17R variant in complex with the increased-selectivity target KLK6 and the reduced-selectivity target mesotrypsin. We were able to obtain a high-resolution crystal structure of the APPI-3M-T11V/G17R variant bound to mesotrypsin (PDB ID:

6GFI; Supplementary Table 3). A structural analysis of this complex revealed that a deleterious steric interaction between the APPI Arg-17 mutation and mesotrypsin Arg-193 pushes Arg-193 into a more buried conformation (Supplementary Figure 8), as previously found in the structures of mesotrypsin bound to wild-type APPI or BPTI Kunitz-type inhibitors[53,54]. The steric clash and the restriction of Arg-193 to a single buried conformation can explain the reduction in affinity toward mesotrypsin, which is consistent with our prior structure and mutagenesis studies[51,55]. The corresponding amino acid that occupies position 193 in KLK6 is Gly (PDB ID: 4D8N); therefore, the lack of a side chain in position 193 of KLK6 is probably more energetically favored (upon binding to APPI-3M-G17R) than that of mesotrypsin Arg-193 (due to the steric clash and the restriction of Arg-193). Efforts to crystallize the APPI-3M-T11V/G17R complex with KLK6 were unsuccessful, and thus the basis for selectivity improvements toward this alternative target, and for cooperativity between APPI residues 11 and 17, remain a subject for future investigations.

**Selective screens are superior to affinity screens**. A significant advantage of our pairwise selectivity screen approach over the traditional sequential affinity screen (a commonly used method, in which the library is sorted against each enzyme separately in a sequential manner[40,43]) is the ability of our approach to identify, in a single screening step (rather than two sequential affinity screen steps), the top ~5% of clones that are more selective toward one target versus another, even if the absolute affinity of these clones toward both targets is lower than that of the parent variant (in the current study, APPI-3M). To demonstrate that the traditional sequential affinity screens are unable to detect the clones obtained by our pairwise selectivity screens (namely, those with improved selectivity and low affinity), we performed two separate sequential affinity screens, one toward KLK6 and another toward cationic trypsin (Supplementary Figure 9A, B, D, E). As expected, both the sequential affinity and the pairwise selectivity screen approaches were able to identify the G17R mutation as a KLK6 selectivity-improving mutation (Supplementary Table 4), which is consistent with the 1.7-fold improvement in the selectivity toward KLK6 versus cationic trypsin, measured by the enzymatic assay (Supplementary Table 5). In contrast, we were unable to identify the selective G17E mutation by using the sequential affinity approach (Supplementary Table 4), although it was clearly identified using the pairwise selectivity screen between KLK6 and cationic trypsin (Supplementary Table 4), demonstrating a 3.4-fold improved selectivity toward cationic trypsin, as measured by the enzymatic assay (Supplementary Table 5). This discrepancy between the two approaches stems from the fact that the G17E mutation was not in the top ~5% binders in the cationic trypsin and KLK6 sorts due to its weakened affinity toward cationic trypsin and KLK6 relative to the parental molecule APPI-3M (by ~4-fold and ~10-fold, respectively, Supplementary Table 5).

**Upscaling**. Our selective pairwise screening approach can be easily scaled up for multiple target proteins per screen, such that a library

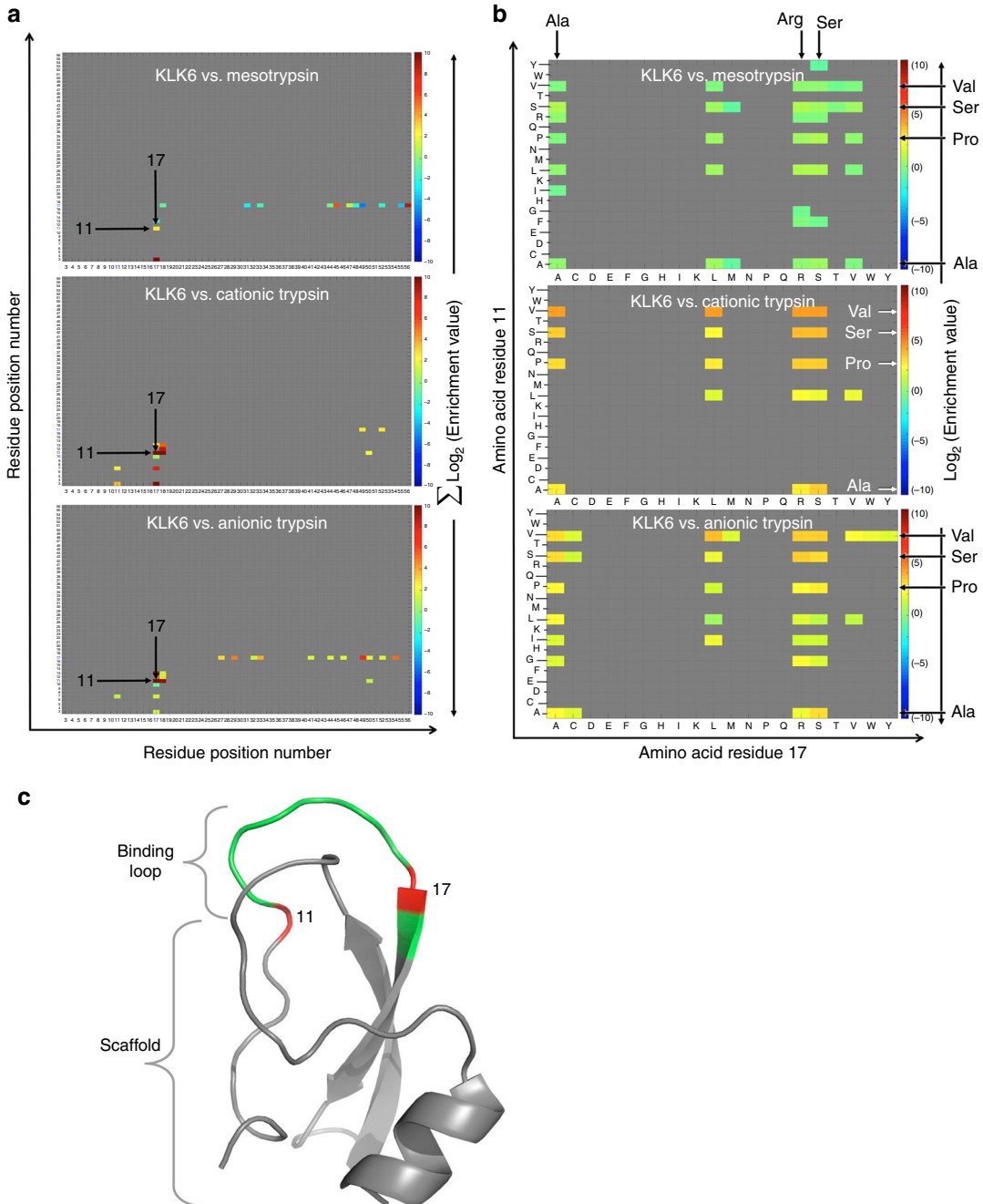

**Fig. 3** Double mutation selectivity landscape of APPI-3M. Heatmaps demonstrating the effect of double-amino acid mutations in APPI-3M on the selectivity toward KLK6 versus the three other serine proteases. **a** The effect of different pairs of mutated residues on selectivity is illustrated by the colors of the heatmaps (red = increased selectivity, enrichment ratio >1; blue = decreased selectivity, enrichment ratio <1). The contribution of each double mutation to selectivity was summed and the maps demonstrate the overall effect. The *X* and *Y* axes indicate the position of the substituted amino acid residues. See Supplementary Figure 4 for further details. **b** The effect of different amino acid mutations at residues 11 and 17 of APPI-3M on its selectivity toward KLK6, illustrated by the colors in the heatmaps. The *X* axis indicates amino acids mutated at residue 17 and the *Y* axis indicates amino acids mutated at residue 11. **c** Crystal structure of APPI-3M (PDB ID: 5C67). Cartoon representation of APPI-3M illustrating the positions of correlated residues Thr-11 and Gly-17 (red) within the APPI binding loop (green, positions 11–18). The APPI scaffold is shown in gray

can be screened against a target of interest (labeled with one type of fluorophore) versus a mixture of competitors (all labeled with the same fluorophore, which is different from the one used for the target of interest). Such an approach is especially useful in the case where there is a single primary target of interest, since it will be completed through only a single sort. To demonstrate the feasibility of such an approach, we performed a competitive sort, in which KLK6 was the primary target of interest (labeled with Alexa Fluor-650) and cationic trypsin, anionic trypsin, and mesotrypsin (each

labeled with Alexa Fluor-488) were the competitors (Supplementary Figure 9C, F), and compared the enrichment values to those of our pairwise comparisons. The enrichment ratios of the competitive multi-target screen were highly correlated with those of the pairwise selective screen; in both setups, the top-rated selectivity-improving clones were similar (Supplementary Table 6), both for single mutations (e.g., G17R) and for double mutations (e.g., T11S/G17R and T11V/G17R). In addition, this analysis revealed a clear selectivity cold spot, in which most mutations in residue 17 increased the

**Table 2 Changes in the selectivity of APPI-3M variants with mutations at selectivity-switch residues toward human serine proteases**

| Mutation | Target A | Target B | Predicted switch[a] | $K_i$ [pM], target A[b] | $K_i$ [pM], target B[b] | Switch ratio[c] |
|---|---|---|---|---|---|---|
| T11H | Mesotrypsin | Anionic trypsin | A ->B | 302 ± 24 | 4.45 ± 0.23 | **1.6** |
| T11I | | | B ->A | 61.2 ± 2.6[a] | 1.44 ± 0.14 | |
| $K_i$ Fold change[d] | — | — | — | 4.93 | 3.09 | — |
| G17E | KLK6 | Cationic trypsin | A ->B | 464 ± 39 | 85.8 ± 5 .9 | **5.7** |
| G17R | | | B ->A | 77.4 ± 2.6 | 8.12 ± 0.08 | |
| $K_i$ Fold change[e] | — | — | — | 59.95 | 10.57 | — |

[a]As predicted by the NGS analysis
[b]Results (means ± SD) were obtained from three independent experiments
[c]Calculated as the $K_i$ fold change for target A divided by the $K_i$ fold change for target B
Switch ratio = $\frac{K_i \text{ fold for target A}}{K_i \text{ fold for target B}}$
[d]$K_i$ (T11H)/$K_i$ (T11I)
[e]$K_i$ (G17E)/$K_i$ (G17R)

**Table 3 $K_i$ constants of human serine proteases inhibited by various APPI-3M variants**

| Mutant | [a]$K_i$ [pM] | | | |
|---|---|---|---|---|
| | Mesotrypsin | Anionic trypsin | Cationic trypsin | KLK6 |
| Unmodified APPI-3M | 98.0 ± 1.0 | 2.26 ± 0.08 | 22.5 ± 0.6 | 362 ± 10 |
| T11V/G17R | 494 ± 28 | 0.92 ± 0.07 | 2.37 ± 0.17 | 16.4 ± 0.9 |
| T11S/G17R | 1060 ± 30 | 2.98 ± 0.19 | 7.25 ± 0.5 | 124 ± 13 |
| T11S | 581 ± 7 | 1.16 ± 0.09 | 7.63 ± 0.55 | 1000 ± 60 |
| T11V | 65.0 ± 1.0 | 3.72 ± 0.21 | 14.1 ± 0.5 | 378 ± 9 |
| G17R | 676 ± 8 | 3.58 ± 0.16 | 8.12 ± 0.08 | 77.4 ± 2.6 |

[a]Results (means ± SD) were obtained from three independent experiments

binding selectivity toward KLK6 versus all other proteases (Supplementary Figure 3E). This finding is consistent with those obtained using the pairwise screening approach (Supplementary Figure 3B).

## Discussion

We describe a novel strategy for mapping the binding selectivity landscapes of proteins through a combination of experimental multi-target selective library screening and in silico next-generation sequencing analysis. Employing the APPI/serine protease system as a model PPI, we show that our strategy can be used to map, in a rapid, single-step, cost-effective process, several crucial aspects of the selectivity landscape, including hot-spot residues, selectivity switch residues, and correlated-selectivity residues. The latter are of special importance, as characterizing correlated-selectivity mutations and analyzing their effects (both individually and combined) on target affinity and selectivity is challenging with currently available approaches[27,56].

Several previous studies have combined selective screening of a protein library and NGS analyses to map the binding landscape of various proteins, including influenza inhibitors (HB36.4, HB80.3)[7], the human Yes Associated Protein 65 (hYAP65) WW domain[19], and an anti-VEGF antibody[40]. However, these approaches employed either libraries of clones with only single mutations or library screens that were performed against only a single target. Therefore, in these previous studies, it was difficult to identify mutations that change target selectivity or that work in concert to affect target affinity and selectivity in a correlated manner. Thus, a major advantage of our approach is its ability to identify correlated-selectivity mutations. For example, we found that the mutations T11V and G17R, when combined, yield a highly potent and

selective inhibitor for KLK6, while combining the mutations T11S and G17R yield only a moderately potent and partially selective inhibitor for KLK6. These findings may suggest that a small and hydrophobic amino acid (e.g., Val in position 11) exerts a stronger effect on selectivity towards KLK6 than a small and polar amino acid (Ser in position 11).

Another advantage of our approach lies in using a pairwise selectivity screen, rather than the sequential affinity screen that is commonly used in other approaches[40,43], to increase selectivity. This advantage is especially noticeable for the identification of clones that are selective but have distinct affinities toward both targets that are lower than that of the parent variant (in the current study, APPI-3M), as demonstrate in Supplementary Table 4. In addition, the pairwise screening approach can be easily scaled up for multiple target proteins per screen, such that a library can be screened against a target of interest versus a mixture of competitors. Such an approach is especially useful where there is a single primary target of interest, since it will be completed with only one sort, as demonstrate in Supplementary Table 6.

We chose the serine protease family as an ideal group of targets to demonstrate our strategy mainly because inhibiting the human serine proteases is of clinical value: both KLK6 and mesotrypsin are involved in cancer progression[57–59], while anionic and cationic trypsins are involved in the etiology of pancreatitis[60,61]. However, the development of inhibitors capable of discriminating among trypsin-like proteases has been challenging. We and others have previously used X-ray crystallography to explore the structures of these proteases, in some cases identifying the distinguishing features that suggest the potential for developing highly selective inhibitors[54,62,63]. For example, several adaptive mutations have been shown to shape the active site of mesotrypsin for distinct substrate and inhibitor-binding selectivity[54,63–65]. Nevertheless, the development of truly selective inhibitors has yet to be achieved, and we anticipate that our novel approach, which is capable of rapidly and efficiently screening large libraries to comprehensively map selectivity, will enable the development of selective probes and therapeutic agents.

APPI has attracted our interest as a scaffold for engineering selective serine protease inhibitors due to the marked sequence diversity among Kunitz family members, which possess canonical binding loops that are highly tolerant to substitution or incorporation of additional amino acids[66,67]. Because the sequence of the canonical binding loop and neighboring residues largely determine the affinity and selectivity of the inhibitor to its targets[53,68], using APPI as a scaffold offers a unique opportunity to optimize target affinity and selectivity without compromising

**Table 4 The selectivity of APPI-3M variants (normalized to the unmodified APPI-3M) toward KLK6 versus the three other proteases**

| Mutant | vs. mesotrypsin | vs. anionic trypsin | vs. cationic trypsin | Calculated KLK6 total selectivity[a] | Expected KLK6 total selectivity[b] |
|---|---|---|---|---|---|
| Unmodified APPI-3M | 1 | 1 | 1 | 1 | — |
| T11V/G17R | 111.10 | 8.94 | 2.32 | 2304.13 | 242.45 |
| T11S/G17R | 31.62 | 3.87 | 0.94 | 115.53 | 32.33 |
| T11S | 2.14 | 0.30 | 0.12 | 0.08 | — |
| T11V | 0.64 | 1.58 | 0.60 | 0.60 | — |
| G17R | 32.26 | 7.42 | 1.69 | 404.09 | — |

[a]Calculated selectivity = $\dfrac{\frac{Ki_{WT} \text{ for KLK6}}{Ki_{mutant} \text{ for KLK6}}}{\frac{Ki_{WT} \text{ for other protease in the pair}}{Ki_{mutant} \text{ for other protease in the pair}}}$

[b]Expected selectivity of double-mutant AB = calculated selectivity of A × calculated selectivity of B

stability. In addition, the affinity of the complexes between APPI and mesotrypsin, anionic trypsin, and cationic trypsin is similar, which facilitated the identification of cold spots, whereas the affinity of the APPI/KLK6 complex is three orders of magnitude lower than that of the other complexes, thus allowing us to identify selectivity-switch residues.

As a validation of the utility of our platform, we show that the results obtained using NGS of the selected APPI clones typically correlate well with the binding selectivity of the purified protein variants in solution (as measured by competitive inhibition studies), but at different scales (Supplementary Table 7). For example, the selectivity values of 13 combinations of enzyme–inhibitor variants (out of a total of 15 possible combinations examined), calculated using NGS, are well-correlated (whether the selectivity was improved or damaged) with those obtained in the enzymatic assay. Of note, in all 15 combinations, a clear correlation was found between the ranking of the selectivity values that were calculated by each method (ranking is according to the level of selectivity improvement within each method for each enzyme, with the greatest improvement ranked as one; see example in bold boxes in Supplementary Table 7). As shown in Table 1, the NGS analysis predicted a selectivity increase of ~$7 \times 10^3$-fold from cationic trypsin to KLK6 for G17R compared with G17E, and of ~$70 \times 10^3$-fold from anionic trypsin to mesotrypsin for T11I compared with T11H; both these findings are in qualitative agreement with the increase in selectivity determined from the $K_i$ values of the soluble proteins, namely, an increase of ~5.7-fold and ~1.6-fold, respectively (Table 2). However, no correlation was found between the magnitudes of the improvements, i.e., the $7 \times 10^3$-fold improvement calculated by NGS was calculated as a ~5.7-fold improvement in the enzymatic assay, while the $70 \times 10^3$-fold improvement calculated by NGS was calculated as only a ~1.6-fold improvement in the enzymatic assay. Therefore, the selectivity increase values that were calculated by the NGS cannot be directly compared with those of the competitive inhibition studies; rather, the values can be compared between experiments using each method, and not between the two methods. Nevertheless, the results shown in Tables 1 and 2 confirm that our approach can predict the positions that can change target selectivity, and that our approach is sufficiently sensitive to detect small affinity changes, whereas other currently available approaches can typically identify only greater changes in the interactions between proteins[69].

In further validation of our strategy, we identified most previously described mutations that affect the binding affinity and selectivity of APPI to serine proteases, as well as some novel mutations. For example, we identified residue 15 as a hot spot for all four human serine proteases, as all amino acid mutations in this residue (except R15K) reduced the binding affinity of APPI-3M to each of the four proteases (Fig. 2 and Supplementary Figure 3). Indeed, residue 15 had previously been identified as a hot spot in Kunitz-domain

inhibitors in studies with BPTI[70,71]. In addition, our data identified, for the first time, to the best of our knowledge, that residue 13 is a selectivity cold-spot for mesotrypsin, as most of the mutations in this residue improved selectivity toward mesotrypsin versus all other proteases. On the other hand, mutating the residue in position 11 switched the selectivity from anionic trypsin to mesotrypsin. Therefore, the difference between residues 13 and 11 is that the former facilitates a selectivity switch from three proteases to a specific protease, while the latter enables a selectivity switch from one protease to one other protease.

The use of NGS covered the entire library and provided a comprehensive map of the binding interface. However, generating the library by using a combination of site-specific saturation mutagenesis on the APPI loop, and random mutations also on other parts of the gene, limited our ability to analyze residues that are distant from the interaction site. We attribute this limitation to technical aspects of our library design, as the random mutations generated by using the error-prone PCR were represented to a lower extent than mutations generated by using site-saturation libraries. Nevertheless, the residues that we found to improve the selectivity of APPI toward the four serine proteases can provide an explanation for the basis for target selectivity of inhibitors toward serine proteases. These selectivity-improving mutations can also be beneficial for designing targeted therapeutics for cancer and other diseases, as they can potentially inhibit the desired serine protease in a selective manner, so as to minimize toxic effects. This study also serves as an example for the general utility of our new platform, as many PPI mediators and disease targets belong to large families of related proteins, making target selectivity a highly desirable but challenging goal in drug development. Thus, we our approach for simply and efficiently mapping PPI selectivity landscapes offers great promise for designing novel target-selective therapeutics.

## Methods

**YSD and flow cytometry cell sorting**. The yeast-displayed APPI-3M library was constructed as described in Supplementary Methods. To display the APPI-3M library on the surface of the yeast, the library was grown in an SDCAA selective medium (2% dextrose, 0.67% Difco yeast nitrogen base, 0.5% Bacto casamino acids, 0.52% Na$_2$HPO$_4$, and 0.856% NaH$_2$PO$_4$·H$_2$O) and induced for expression with a galactose medium (as for SDCAA, but with galactose 2%, instead of dextrose) according to an established protocol[72]. Inactive forms of mesotrypsin, anionic trypsin, and cationic trypsin containing the mutation S195A were used as a precaution against enzymatic cleavage during the experiments[51]. The four serine proteases were labeled with Alexa Fluor dyes (Invitrogen, Carlsbad, CA) and used to detect binding. For pairwise selectivity screen, ~ $1 \times 10^8$ yeast cells were incubated with different Alexa Fluor-labeled serine proteases in a binding buffer (100 mM Tris, pH = 8.0, 1 mM CaCl$_2$, 1% BSA) for 1.5 h at room temperature. Then, the cells were washed with the binding buffer and sorted for the high-selective variants by conducting several independent sorts, using FACSAria [the Ilse Katz Institute for Nanoscale Science and Technology, Ben-Gurion University of the Negev (BGU), Israel]. The complexes included the following pairs and

concentrations: mesotrypsin/KLK6 [25 nM/7 nM], mesotrypsin/cationic trypsin [25 nM/8 nM], mesotrypsin/anionic trypsin [25 nM/12.5 nM], anionic trypsin/ KLK6 [250 nM/7 nM], cationic trypsin/KLK6 [100 nM/7 nM], and anionic trypsin/ cationic trypsin [12.5 nM/100 nM]. APPI-3M variants that showed a high binding affinity (top 5% of the entire population) toward one serine protease in the pair and a low binding affinity toward the other were selected. Dual-color flow cytometry (BD Accuri C6, Piscataway, NJ) was used to test the selective binding of each sorted library to one serine protease in the pair in the presence of the other.

**Quality filtration and integration of sequences**. Sequencing data from each library were treated identically and evaluated in triplicates, and Spearman's rank correlation coefficient[73] was calculated to be above 95%. An average Illumina quality score was calculated for each read in a given set of paired-end reads, and read pairs in which either read had an average quality score lower than 20 (i.e., less than 99% accuracy) were discarded. The remaining read pairs were merged into a single sequence by fast length adjustment of short reads (FLASH) software[74]. DNA sequences and their amino acid translations were aligned to the sequence of APPI-3M; sequences of different lengths and sequences containing stop codons were discarded.

**Computational analysis of high-throughput sequencing results**. The analysis was performed in MATLAB, version R2016a. Variants with one amino acid mutation and variants with multiple amino acids mutations were analyzed separately. First, the number of reads of each variant from each library was counted. Then, to avoid variants with a low number of reads (which can yield noisy frequencies and enrichment ratios), we determined a threshold value of 100 reads for variants with a single amino acid mutation and 10 reads for variants with double amino acids mutations. Variants with read numbers below the threshold in the naive and sorted library fractions were discarded, and variants with read numbers below the threshold value received the threshold value if the read number of the variant in the other library was above the threshold.

Next, the frequency of each remaining variant, v, from each library was computed as $F_v = \frac{\text{Reads}_v}{\sum \text{Reads}_v}$, where $\text{Reads}_v$ is the number of times that this variant appeared in the library. Based on its frequency, the enrichment ratio of each variant from each sorted library was calculated. The enrichment ratio for a given variant, v, was calculated as $ER_v = \frac{F_{v,\text{sorted}}}{F_{v,\text{naive}}}$, where $F_{v,\text{sorted}}$ is the frequency of the variant in the sorted library and $F_{v,\text{naive}}$ is the frequency of the same variant in the naive (pre-sorted) library. Eventually, for single amino acid substitution, heatmaps were created based on the enrichment ratio[5]; for double amino acid substitutions, we summed the enrichment ratios of similar double-mutation variants that have mutations in the same residues ($\sum ER_{x,y} = ER_1 + ER_2 + \ldots + ER_N$, where x and y are the mutated residues and N is the number of substitutions at the x and y residues). We illustrated these results as heat maps.

## Data availability

All relevant data are available from the authors. The coordinates and structure factors for the complex of APPI-3M-T11V/G17R variant bound to mesotrypsin have been submitted to the Protein Data Bank (PDB) under the accession code 6GFI. The crystal structure of APPI-3M is available in the PDB under the accession code 5C67. The crystal structure of KLK6 is available in the PDB under the accession code 4D8N. The crystal structure of the mesotrypsin/BPTI complex is available in the PDB under the accession code 2R9P.

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

## Acknowledgements

The authors thank Vered Caspi (BGU), Matan Shemer (BGU), and Jonathan Barlev (Weizmann Institute of Science, Israel) for their helpful discussions. We thank Dr. Uzi Hadad for his technical assistance. FACS experiments were performed at the Ilse Katz Institute for Nanoscale Science & Technology, BGU. N.P. acknowledges support from the European Research Council "Ideas program" ERC-2013-StG (contract grant number: 336041). N.P. and E.S.R. acknowledge support from the US-Israel Binational Science Foundation (BSF). E.S.R. acknowledges support from the United States National Institutes of Health grant number R01CA154387. The structural studies were performed on beamline ID30-B at the European Synchrotron Radiation Facility (ESRF), Grenoble, France. We are grateful to Christoph Mueller-Dieckmann for providing assistance in using this beamline. We would like to thank Prof. Kay Diederichs and Dr. Ronan Keegan for their help and contribution in the structure determination during the 1st CCP4/BGU Structure Solution Workshop, which took place at Ben-Gurion University of Negev during February 2018.

## Author contributions

S.N and I.C. made an equal contribution as first authors; S.N., I.C., and N.P. designed the research; S.N., I.C., A.H., and E.S.R generated the proteins; S.N. and I.C. performed the research; S.N., I.C., A.S., E.S.R., and N.P. analyzed the data; S.N., I.C., and N.P. wrote the paper. All authors edited the manuscript and approved the final version.

## Additional information

**Competing interests:** The authors declare no competing interests.

