## [Peer Review File · Nature Communications]

Reviewers' comments:

Please note that Reviewer 3 is our bioinformatics and computational data analysis expert reviewer.

Reviewer #1 (Remarks to the Author):

In this manuscript, the authors describe a procedure for using library screening with next-gen sequencing (NGS) to find protein variants that exhibit enhanced selectivity. Using this approach, mutations in the APPI-3M protease inhibitor are identified that have altered specificity for binding 4 target proteases.

The manuscript claims novelty from this procedure, however these claims are somewhat overstated. Previous methods have typically screened against multiple targets sequentially, then retrospectively analyzed the data to identify mutants with the desired binding profile. In contrast, here the authors instead screen with two labeled targets, such that the screen directly yields selective clones. This is a nice, and elegant, idea. However, it is not really clear why this method is superior to the "traditional" approach of screening against each target sequentially. Perhaps more accuracy is gained when assigning differences in binding to clones that are much weaker than the starting point? And thus, this method works better in the regime where one seeks a selective binder, and not a variant that will bind to many targets at once (as Jardine and Koenig were looking for)? Regardless, there is no real demonstration of the advantage of this method over screening sequentially, and this should be included if the authors wish to claim utility of their method.

In fact, since this method treats all targets in a pairwise exhaustive way (though in future this does not need to be the case), it scales poorly. Given 29 targets (as in the Jardine paper cited here), one would much rather screen the set once each, rather than having to consider each of the $29 \times 28 / 2 = 406$ possible pairs separately.

Regarding the data arising from the screen itself, it is not clear that the authors are truly selecting on the basis of binding affinity, and not association rate. Based on the very tight binding constants of the WT construct, the off-rates must be very very slow. This is problematic because if the labeled proteases are not allowed enough time to dissociate, they simply report on the first inhibitor they bind to: thus, this experiment would actually report on association kinetics, and not binding affinity. The binding partners are incubated for 1 hour prior to sorting, and there is no evidence presented to show that equilibrium has been reached before sorting takes place.

Also, the authors claim agreement between the NGS data and the enzyme assay that is far overstated. The NGS data tells us that His vs Ile at position 11 leads to a 69,000-fold difference between meso and anionic trypsin. But then the corresponding enzyme assay, setup to yield an analogous metric, gives only a 1.6-fold difference. These are very different! Beyond this, only two positions are shown (residues 11 and 17), but the order of the selectivity difference is opposite for the two methods: NGS gives a bigger selectivity difference at position 11, and the enzyme assay gives a bigger effect at position 17. Obviously it's critical to validate the NGS data with a separate binding experiment (and this enzyme assay is suitable to do so). But instead of validating the NGS results, here the results from the enzyme assay don't really seem to agree much at all.

Additional points:

- 1) Having identified these mutants, it would be really helpful to go back to the structures of the complexes, to try and rationalize how they're working. Even if it's not obvious, that's fine too. But at least showing where these mutations are in the complex and what the WT interaction looks like is easy to do and would be helpful for the reader.
- 2) The descriptions of hotspots / coldspots / specificity-switches / etc. in the introduction are not

very clear, and will not help an uninitiated reader to actually understand these ideas.

3) The axis labels of Figure 2 (and Figure 3) are not legible. More important, the labels on the heatmap scale at the far right is not legible, this means that it's impossible to actually see the magnitude of the effects on these heatmaps.

4) In Figure S3, the protein elutes from the sizing column in two peaks. Why is this?

5) In Figure S5, numbers are presented for the double-mutant cycles, but these are not actually explained.

Reviewer #2 (Remarks to the Author):

This manuscript by Naftaly, et al. lays out a powerful and general approach to mapping out residues important in determining the selectivity of binding of a given protein to two or more different binding partners. This is a topic of some interest, particularly for protein engineers interested in developing selective protein inhibitors of other factors. Using a non-selective trypsin inhibitor (APPI) as a starting point, these workers made a comprehensive set of mutations in the (known) binding loop of the protein as well as carried out mutagenic PCR to sprinkle mutations into other regions of the protein as well. They then displayed this library of APPI mutants on the surface of yeast cells and incubated the cells with red- and green-labeled derivatives of two proteins with which APPI interacts. They then use FACS to isolate yeast that display APPI variants that are enriched or discriminated against relative to their representation in the original library. This would indicate that they have some effect on the selectivity of the APPI for one of the target proteins with respect to the other. The power of deep sequencing makes this comprehensive analysis possible.

They succeed in identifying interesting mutants that very clearly alter the selectivity of APPI relative to the starting point. Particularly impressive is the fact that they identify double mutants that when mutated in tandem, grossly alter the selectivity of the protein. This illustrates the power of comprehensive library coverage using this FACS-based technique and deep sequencing.

The conclusions derived from the high-throughput screening data were validated by expressing individual mutants and characterizing their ability to inhibit the activity of individual tryptic protease partners. The results corroborated the screening data nicely.

In general, this is a nice study that should be of interest to protein engineers interested in discovering altered selectivity mutants. I have only a couple of minor suggestions.

First, the authors may wish to replace the word "specificity" with "selectivity". The former is a kind of absolute term, whereas the second is relative and thus more appropriate.

Second, they should acknowledge that this approach to screen libraries for molecules that bind selectively to one protein over another has been reported by Mendes, et al. last year.

Mendes, K., Malone, M.L., Ndungu, J.M., Saponitsky-Kroyter, I., Cavett, V., McEnaney, P.J., MacConnell, A.B., Doran, T.M., Ronacher, K., Stanley, K., Utset, O., Walzl, G., Paegel, B.M. and Kodadek, T. (2017) "High-throughput identification of DNA-encoded IgG ligands that distinguish active and latent Mycobacterium Tuberculosis infections" ACS Chem. Biol. 12, 234-243.

These workers used a DNA-encoded library of bead-displayed library of synthetic molecules rather than a yeast-displayed library of protein mutants, but the approach is nearly identical. Mendes, et al. also labeled targets with a red dye and off-targets with a green dye and then used FACS to sort beads that were enriched for red over green. The DNA tags were amplified and deep sequenced to reveal the nature of the selective ligands. The authors probably missed this paper since it is in the

chemical literature.

It is important to acknowledge this precedent as far as assay development, but it is quite far removed from this type of application, so it should not detract from the impact of this study as a tool for protein engineering.

Reviewer #3 (Remarks to the Author):

The study shows a new technique for accurate characterizing of binding specificity landscape of protein-protein interactions (PPI). The authors tested their techniques on an example of binding of the amyloid protein precursor inhibitor (APPI) to each of four human serine proteases.

The strategy consists of 4 main steps:

- 1) generating a library of inhibitor mutants by error-prone PCR;
- 2) measuring the binding affinity by experimental multi-target selective library screening (the authors used Yeast-surface display (YSD) which recommended itself in previous studies as a reliable tool to detect the changes in binding affinity);
- 3) determining mutant sequences by NGS;
- 4) building specificity landscape and analysis.

The new technique looks universal and can be potentially used for accurate investigation of landscapes of other PPIs.

During the study, the authors have found several mutations which affect binding strength more than others. They separated mutations into three groups: hotspots - mutations which decrease specificity to protease; cold spots - mutations which increase the specificity to protease; and switches - mutations which change specificity from one protease to another. Naftaly et al. isolated mutations T11H, T11I, G17E, G17R, which were switches and investigated their cumulative effect.

Overall, I would recommend the paper for publications after addressing the comments below.

The primary concern about the study is that its results look narrow. The authors acknowledged that there are many papers about dependence between PPI and mutations. But all of them are concentrated only on single mutations, while the authors' new approach investigates binding landscape with reliance on multiple mutations.

Nevertheless, the paper presents an investigation of only one pair of mutations as an example. And it is not clear why they choose precisely this pair. Was it a random choice or the result of the analysis? Why there is only one example but no three or five or more? How is scalable and practical the method if it can investigate just one pair?

Minor concerns:

- 1) The authors provided the Figure 3 which supposed to demonstrate switch effect of all possible pairs of mutations. The picture shows a table where colors present the outcome of mutations. The color is a summed effect of all possible mutations. Why was summation chosen? Why not min or max? It is better to be explained because different mutations at the same position can have a opposite effect, which is shown in Figure 2.
- 2) What is fluorescence-activated cell sorting (FACS)? (p.5) Should it have a reference?
- 3) Figures 2 and 3 have small labels. I would leave the figures as it is but created enlarged copies of them in supplements with better quality.
- 4) Figure S5 B does not prove the described statement that T11S/G17R are working in tandem, rather T11S decrease specificity. (p. 12)

In general, while the study and introduced technique have huge potential, the report looks raw. The authors gave abrupt quantitative analysis and did not provide a broad comparison between single vs. double mutations effect.

The authors concentrated their attention describing an only specific example, leaving the general analysis of mutation landscape without consideration. The analytics in the study should be entirely reorganized, or it should be explained why the only one example is worth to be published.

Reviewer #1:

In this manuscript, the authors describe a procedure for using library screening with next-gen sequencing (NGS) to find protein variants that exhibit enhanced selectivity. Using this approach, mutations in the APPI-3M protease inhibitor are identified that have altered specificity for binding 4 target proteases.

Q1

The manuscript claims novelty from this procedure, however these claims are somewhat over-stated. Previous methods have typically screened against multiple targets sequentially, then retrospectively analyzed the data to identify mutants with the desired binding profile. In contrast, here the authors instead screen with two labeled targets, such that the screen directly yields selective clones. This is a nice, and elegant, idea. However, it is not really clear why this method is superior to the "traditional" approach of screening against each target sequentially.

Q1.1 Perhaps more accuracy is gained when assigning differences in binding to clones that are much weaker than the starting point?

Q1.2 And thus, this method works better in the regime where one seeks a selective binder, and not a variant that will bind to many targets at once (as Jardine and Koenig were looking for)?

Q1.3 Regardless, there is no real demonstration of the advantage of this method over screening sequentially, and this should be included if the authors wish to claim utility of their method.

Answer to Q1 (AQ1)

Advantage of pairwise selectivity screen vs. sequential affinity screen to gain selectivity—we predict that the pairwise selectivity screen would enable us to identify in a single step (compared to two steps for a sequential affinity screen) the ~5% of clones that are more selective for target A relative to target B, even if their absolute affinity towards both targets is lower than that of the starting point (parental) protein variant (e.g., APPI_3M in our work). In fact, as we now show in new experiments and as described below, this is *the only way* to identify clones for which selectivity is due not only to affinity enhancement toward a specific target, but also, importantly, to reduction in affinity toward an alternate target.

AQ1.1. Indeed, our approach is more accurate than traditional approaches because, unlike traditional sequential affinity screens, our approach sorts both low- and high-affinity variants.

AQ1.2. Indeed, our approach, in which both low and high affinity binders are identified, is advantageous over traditional approaches when seeking a selective binder.

AQ1.3. As suggested by the reviewer, in the revised manuscript (pages 16 and 18 written in track changes mode) we performed two separate sequential affinity screens, one toward KLK6 and the other toward cationic trypsin (Fig. S6A, B, D, E). In each of these screens, we sorted the variants with the highest (top 5%) affinity toward the selected protease. As suggested above, we predicted that by performing sequential affinity screens, we would miss variants with improved selectivity but weaker affinity that could only be identified by our pairwise selectivity screens, since these desired variants would not be in the top 5% in terms of absolute affinity. We also predicted that both the pairwise selectivity and sequential affinity screens would allow us to identify the subset of selective clones that are in the top 5% in terms of absolute affinity.

Page 16: “As expected, both the sequential affinity and the pairwise selectivity screen approaches were able to identify the G17R mutation as a KLK6 selectivity improved mutation (Table S3), which is consistent with the 1.7-fold improvement in the selectivity toward KLK6 versus cationic trypsin as measured by the enzymatic assay (Table S2). In contrast, we were unable to identify the selective G17E mutation by using the sequential affinity approach (Table S3), although it was identified clearly using the pairwise selectivity screen between KLK6 and cationic trypsin (Table S3), demonstrating a 3.4-fold improved selectivity toward cationic trypsin as measured by the enzymatic assay (Table S2). This discrepancy between the two approaches stems from the fact that the G17E mutation was not in the top ~5% binders in the cationic trypsin and KLK6 sorts due to its weakened affinity toward cationic trypsin and KLK6 relative to the parental molecule APPI_3M (by ~4-fold and ~10-fold, respectively, Table S2).”

Q2

In fact, since this method treats all targets in a pairwise exhaustive way (though in future this does not need to be the case), it scales poorly. Given 29 targets (as in the Jardine paper cited here), one would much rather screen the set once each, rather than having to consider each of the $29 \times 28 / 2 = 406$ possible pairs separately.

Answer to Q2

A setup similar to the one presented in the manuscript can be scaled-up to screen multiple competitive targets in a single sort, as we now demonstrate in a new experiment included in the revised manuscript. In such a setup, the library is screened against the target of interest (labeled with one fluorophore) versus a mixture of all other competitors (all labeled with the same fluorophore, which is different from the fluorophore used for the target of interest). Given 29 targets, for example, the target of interest will be labeled with one fluorophore and all other 28 targets/competitors will be labeled with another fluorophore. This multiple competition screen could enable us to identify in a single sort (compared to the 29 sorts needed using the traditional affinity maturation setup) the ~5% of clones that are more selective for the target of interest than for the competitors. Such a setup is especially practical when a single primary target of interest needs to be screened versus various other competitors, since the screen will be completed with only a single sort. We now emphasize this advantage in the revised manuscript, as follows:

Page 17 (in 'show markup' review mode): “Our selective pairwise screening approach can be easily scaled up for multiple target proteins per screen, such that a library can be screened against a target of interest (labeled with one type of fluorophore) versus a mixture of competitors (all labeled with the same fluorophore, which is different from the one used for the target of interest). Such an approach is especially useful in the case that there is a single primary target of interest since it will be completed with only one sort. To demonstrate the feasibility of such an approach, we performed a competitive sort, in which KLK6 was the primary target of interest (labeled with Alexa Fluor-650) and cationic trypsin, anionic trypsin, and mesotrypsin (all labeled with Alexa Fluor-488) were the competitors (see new Fig. S6C, F), and compared the enrichment values to those of our pairwise comparisons. The enrichment ratios of the competitive multi-target screen were highly correlated with those of the pairwise selective screen; in both setups, the top-rated selectivity improvement clones were similar (see new Table S4), both for single mutations (e.g., G17R) or for double mutations (e.g., T11S_+ G17R and T11V+_G17R). In addition, this analysis revealed a clear selectivity cold-spot, in which most mutations in residue 17

increased the binding selectivity toward KLK6 versus all other proteases (see new Fig. S8E). This finding is consistent with the findings obtained using the pairwise screening approach (see new Fig. S8B)."

Q3

Regarding the data arising from the screen itself, it is not clear that the authors are truly selecting on the basis of binding affinity, and not association rate. Based on the very tight binding constants of the WT construct, the off-rates must be very very slow. This is problematic because if the labeled proteases are not allowed enough time to dissociate, they simply report on the first inhibitor they bind to: thus, this experiment would actually report on association kinetics, and not binding affinity. The binding partners are incubated for 1 hour prior to sorting, and there is no evidence presented to show that equilibrium has been reached before sorting takes place.

Answer to Q3

We thank the reviewer for raising this important issue. Unfortunately, by an honest mistake, the original manuscript specified a 1-h incubation time, whereas the actual incubation time was 1.5 h; we are sorry for this mistake, which we have now corrected in the revised manuscript. Based on our previous experience, we expected that 1.5 h would be sufficient time to reach equilibrium using the protein concentrations employed. To confirm that we are indeed selecting based on binding affinity (in equilibrium), we have now performed a FACS kinetic experiment in which we measured the fluorescence intensity of binding for each protease to the yeast cells at different incubation times (using the same protease concentrations used in the manuscript). The results of this experiment (see figure below) indicate that a 90 min incubation is sufficient for all examined proteases to reach >90% of equilibrium, demonstrating that that our sortings were performed near equilibrium.

Q4

Also, the authors claim agreement between the NGS data and the enzyme assay that is far overstated. The NGS data tells us that His vs Ile at position 11 leads to a 69,000-fold difference between meso and anionic trypsin. But then the corresponding enzyme assay, setup to yield an analogous metric, gives only a 1.6-fold difference. These are very different! Beyond this, only two positions are shown (residues 11 and 17), but the order of the selectivity difference is opposite for the two methods: NGS gives a bigger selectivity difference at position 11, and the enzyme assay gives a bigger effect at position 17. Obviously

it's critical to validate the NGS data with a separate binding experiment (and this enzyme assay is suitable to do so). But instead of validating the NGS results, here the results from the enzyme assay don't really seem to agree much at all.

Answer to Q4

We address this point in detail in the revised manuscript (see page 19 and new Table S5):

"As a validation of the utility of our platform, we show that the results obtained using NGS of the selected APPI clones typically correlate well with the binding selectivity of the purified protein variants in solution (as measured by competitive inhibition studies), but in different scales (Table S5). For example, the selectivity values of 13 combinations of enzyme–inhibitor variants (out of a total of 15 possible combinations examined) calculated using NGS are well-correlated (whether the selectivity was improved or damaged) with those obtained in the enzymatic assay. Of note, in all 15 combinations, a clear correlation was found between the ranking of the selectivity values that were calculated by each method (ranking is according to the level of selectivity improvement within each method for each enzyme, with the greatest improvement ranked as one; see example in bold boxes in Table S5). As shown in Table 1, the NGS analysis predicted a selectivity increase of $\sim 7 \times 10^3$ -fold from cationic trypsin to KLK6 for G17R compared with G17E, and of $\sim 70 \times 10^3$ -fold from anionic trypsin to mesotrypsin for T11I compared with T11H; both these findings are in a qualitative agreement with the increase in selectivity determined from the K_i values of the soluble proteins (namely, an increase of ~ 5.7 -fold and ~ 1.6 -fold, respectively; Table 2). However, no correlation was found between the magnitudes of the improvements, i.e., the 7×10^3 -fold improvement calculated by NGS was calculated as a ~ 5.7 -fold improvement in the enzymatic assay, while the 70×10^3 -fold improvement calculated by NGS was calculated as only a ~ 1.6 -fold improvement in the enzymatic assay. Therefore, the selectivity increase values that were calculated by the NGS cannot be directly compared with those of the competitive inhibition studies; rather, the values can be compared between experiments using each method, and not between the two methods. Nevertheless, the results shown in Tables 1 and 2 confirm that our approach can predict the positions that can change target selectivity, and that our approach is sufficiently sensitive to detect small affinity changes, whereas other currently available approaches can typically identify only greater changes in the interactions between proteins".

Additional points:

Q5

1) Having identified these mutants, it would be really helpful to go back to the structures of the complexes, to try and rationalize how they're working. Even if it's not obvious, that's fine too. But at least showing where these mutations are in the complex and what the WT interaction looks like is easy to do and would be helpful for the reader.

Answer to Q5

We agree with the reviewer that a spatial representation of the mutations followed by a structural analysis will be helpful for the reader. As for spatial representation of the mutations we added a new panel for Figure 3 (Fig. 3C), illustrating the positions of the correlated residues Thr-11 and Gly-17 on the parental APPI_3M scaffold. In line with the above comment, and since we believe that showing the crystal structure of the complex that includes the mutations is even more informative than showing the mutations superimposed on the scaffold of the parental (APPI_3M) protein, we attempted to crystallize

complexes of the APPI-3M-T11V/G17R variant, which showed increased selectivity toward KLK6 and reduced selectivity toward mesotrypsin. We were able to obtain a high-resolution crystal structure for the APPI-3M-T11V/G17R variant bound to mesotrypsin (PDB ID: 6GFI; new Table S6 and new Fig. S7). The analysis of the structure of this complex is now presented in the main text (methods and results sections) and in the SI (methods- page 27; results- page 15; SI page 36). Unfortunately our parallel efforts to crystallize the APPI-3M-T11V/G17R complex with KLK6 were less successful, but nevertheless the new structure provides insight into the reduced selectivity of this variant toward mesotrypsin.

Q6

2) The descriptions of hotspots / coldspots / specificity-switches / etc. in the introduction are not very clear, and will not help an uninitiated reader to actually understand these ideas.

Answer to Q6

We shortened and rephrased this part to better clarify these ideas (page 2):

"*Hot-spot residues* are a few⁸ interface residues that are highly relevant for a specific PPI, i.e., they contribute almost 75% of the total free energy of binding ($\Delta\Delta G_{\text{bind}}$) of the protein to its partner⁹⁻¹¹. Mutating hot-spot residues, therefore, decreases the affinity of the protein to a specific partner – but not necessarily to others. *Cold-spot residues*^{1,12,13,14} are interface residues occupied by suboptimal amino acids, such that mutating them increases the binding affinity of the protein to a specific partner. *Selectivity-switch residues*^{15,16} are interface residues in which a point-mutation simultaneously decreases the affinity of the protein to one partner and increases its affinity to another. Finally, *correlated-selectivity residues*^{17,18} are interface residues that work together to increase the selectivity of the protein to one specific partner. Such residues are especially difficult to characterize with conventional methods, because only a double-mutation (one mutation in each residue) can change the affinity of the protein to a certain partner."

Q7

3) The axis labels of Figure 2 (and Figure 3) are not legible. More important, the labels on the heatmap scale at the far right is not legible, this means that it's impossible to actually see the magnitude of the effects on these heatmaps.

Answer to Q7

We agree with the reviewer and generated an enlarged view of Fig. 2 (which shows the binding loop residues) and included it in the SI (see new Fig. S8). We also enlarged the symbols of the amino acids and the labels on the heatmap scale in Fig. 3B, and added an enlarged view of Fig. 3A in the SI (see new Fig. S9).

Q8

4) In Figure S3, the protein elutes from the sizing column in two peaks. Why is this?

Answer to Q8

The peak that corresponds with the protein in Fig. S3B is the left (higher) peak, as was validated by mass spectrometry (data not shown). The smaller peak on the right of the panel does not correspond with the typical elution volume of proteins with a MW such as that of APPI. Considering an elution volume of ~120 ml (which is also the volume of the column) for the right peak, it corresponds with the elution volume of a small molecule, presumably the imidazole that was used for elution from the nickel column. We clarified this issue in the caption of Fig. S3.

Q9

5) In Figure S5, numbers are presented for the double-mutant cycles, but these are not actually explained.

Answer to Q9

We clarified this issue in the revised paper (pages 13, 34 and new Fig. S5). Also, in order to describe the results better (and as a response to Q16) we changed the energy calculations from affinity to selectivity (see new Fig. S5). By doing so, we are now able to show the overall effect of selectivity rather than a specific example of affinity.

Page 13: "These results suggest that residues 11 and 17 are correlated-selectivity residues, which act together to increase target selectivity. To further test this hypothesis, we conducted a double-mutant cycle analysis⁵¹, in which we used the selectivity values of KLK6 with the two double-mutant variants and their single variants (T11V/G17R, T11S/G17R, T11V, T11S, and G17R, Table 4) to calculate the selectivity strength of interactions between two mutated residues (i.e., the coupling energy, $\Delta\Delta G_{int}$; Fig. S5). Indeed, in both double mutations, the $\Delta\Delta G_{int}$ values were non-zero, indicating that residues 11 and 17 interact with each other to co-operatively affect the selectivity toward KLK6."

Page 34, Figure S5 legend: "Fig. S5. Double-mutant cycle analysis for measuring the coupling energy between residues 11 and 17 in KLK6 selectivity. (A) Free energy changes of the T11V/G17R substitution. (B) Free energy changes of the T11S/G17R substitution. To assess whether the effects of the mutations on the measured K_i are independent or correlated (cooperative), we assessed the strength of the interactions between two residues, X and Y, in the protein (P) in a cycle that comprised the wild-type protein P_{XY} , two single mutants, P_{XO} and P_{OY} , and the corresponding double mutant, P_{OO} (O indicates a

mutation). A measure of the strength of the interaction between residues X and Y is considered the

$$\Delta\Delta G_{int} = -RT \ln\left(\frac{S_{XY} \times S_{00}}{S_{0Y} \times S_{X0}}\right)$$

coupling energy, $\Delta\Delta G_{int}$, which is given by:

where R is the gas constant, T is the absolute temperature, and S_X , S_{0Y} , S_{X0} , and S_{00} correspond to the calculated total selectivity. A coupling energy of zero (i.e., additivity of mutational effects) indicates that X and Y do not interact. The free energy changes ($\Delta\Delta G$) upon a single point-mutation (i.e., the $\Delta\Delta G$ of P_{XY} and P_{X0}) were calculated in

$$\Delta\Delta G = -RT \ln \frac{S_{X0}}{S_{XY}}$$

a similar manner, and are given by:

Each ellipse (corners) indicates a different APPI variant, as denoted, and the values near each arrow represent the $\Delta\Delta G$ (kcal/mol). The numbers at the middle of each panel indicate the coupling energy $\Delta\Delta G_{int}$ (kcal/mol)."

Reviewer #2

This manuscript by Naftaly, et al. lays out a powerful and general approach to mapping out residues important in determining the selectivity of binding of a given protein to two or more different binding partners. This is a topic of some interest, particularly for protein engineers interested in developing selective protein inhibitors of other factors. Using a non-selective trypsin inhibitor (APPI) as a starting point, these workers made a comprehensive set of mutations in the (known) binding loop of the protein as well carried out mutagenic PCR to sprinkle mutations into other regions of the protein as well. They then displayed this library of APPI mutants on the surface of yeast cells and incubated the cells with red- and green-labeled derivatives of two proteins with which APPI interacts. They then use FACS to isolate yeast that display APPI variants that are enriched or discriminated against relative to their representation in the original library. This would indicate that they have some effect on the selectivity of the APPI for one of the target proteins with respect to the other. The power of deep sequencing makes this comprehensive analysis possible.

They succeed in identifying interesting mutants that very clearly alter the selectivity of APPI relative to the starting point. Particularly impressive is the fact that they identify double mutants that when mutated in tandem, grossly alter the selectivity of the protein. This illustrates the power of comprehensive library coverage using this FACS-based technique and deep sequencing.

The conclusions derived from the high-throughput screening data were validated by expressing individual mutants and characterizing their ability to inhibit the activity of individual tryptic protease partners. The results corroborated the screening data nicely.

In general, this is a nice study that should be of interest to protein engineers interested in discovering altered selectivity mutants. I have only a couple of minor suggestions.

Q10

First, the authors may wish to replace the word "specificity" with "selectivity". The former is a kind of absolute term, whereas the second is relative and thus more appropriate.

Answer to Q10

We agree with the reviewer and changed the wording throughout, as suggested.

Q11

Second, they should acknowledge that this approach to screen libraries for molecules that bind selectively to one protein over another has been reported by Mendes, et al. last year.

Mendes, K., Malone, M.L., Ndungu, J.M., Suponitsky-Kroyter, I., Cavett, V., McEnaney, P.J., MacConnell, A.B., Doran, T.M., Ronacher, K., Stanley, K., Utset, O., Walzl, G., Paegel, B.M. and Kodadek, T. (2017) "High-throughput identification of DNA-encoded IgG ligands that distinguish active and latent Mycobacterium Tuberculosis infections" ACS Chem. Biol. 12, 234-243.

These workers used a DNA-encoded library of bead-displayed library of synthetic molecules rather than a yeast-displayed library of protein mutants, but the approach is nearly identical. Mendes, et al. also labeled targets with a red dye and off-targets with a green dye and then used FACS to sort beads that were enriched for red over green. The DNA tags were amplified and deep sequenced to reveal the nature of the selective ligands. The authors probably missed this paper since it is in the chemical literature.

It is important to acknowledge this precedent as far as assay development, but it is quite far removed from this type of application, so it should not detract from the impact of this study as a tool for protein engineering.

Answer to Q11

We are sorry for this omission, and we now include this study in the bibliography and address it in the introduction. Page 3. "A more recent approach employed next-generation sequencing (NGS) to guide protein and synthetic small-molecule optimization..."

Reviewer #3

The study shows a new technique for accurate characterizing of binding specificity landscape of protein-protein interactions (PPI). The authors tested their techniques on an example of binding of the amyloid protein precursor inhibitor (APPI) to each of four human serine proteases.

The strategy consists of 4 main steps:

- 1) generating a library of inhibitor mutants by error-prone PCR;
- 2) measuring the binding affinity by experimental multi-target selective library screening (the authors used Yeast-surface display (YSD) which recommended itself in previous studies as a reliable tool to detect the changes in binding affinity);
- 3) determining mutant sequences by NGS;
- 4) building specificity landscape and analysis.

The new technique looks universal and can be potentially used for accurate investigation of landscapes of other PPIs.

During the study, the authors have found several mutations which affect binding strength more than others. They separated mutations into three groups: hotspots - mutations which decrease specificity to protease; cold spots - mutations which increase the specificity to protease; and switches - mutations which change specificity from one protease to another. Naftaly et al. isolated mutations T11H, T11I, G17E, G17R, which were switches and investigated their cumulative effect.

Overall, I would recommend the paper for publications after addressing the comments below.

Q12

The primary concern about the study is that its results look narrow. The authors acknowledged that there are many papers about dependence between PPI and mutations. But all of them are concentrated only on single mutations, while the authors' new approach investigates binding landscape with reliance on multiple mutations.

Nevertheless, the paper presents an investigation of only one pair of mutations as an example. And it is not clear why they choose precisely this pair. Was it a random choice or the result of the analysis? Why there is only one example but no three or five or more? How is scalable and practical the method if it can investigate just one pair?

Answer to Q12

As shown in Fig. 3A, in each pairwise screen of KLK6 we obtained several position pairs within the APPI sequence (at least 13 different pairs as shown in the KLK6/cationic screen, Fig. 3A). The reason for choosing the pair with positions 11 and 17 is because our analysis results showed that this 11&17 pair was most likely to represent correlated residues that improve KLK6 selectivity relative to all other proteins (i.e., KLK6 vs. mesotrypsin, cationic trypsin and anionic trypsin) through a cooperative interaction (Fig. 3A). In addition, positions 11 and 17 showed the highest number of different variants and, therefore, offered a more interesting case study for further analysis as shown in Fig. 3B. In general, the number of identified pairs depends on the investigated system (the proteins used); therefore we believe that for some proteins there would be many possible pairs and for others there would be fewer. In our study, for example, the KLK6/anionic screen provided 21 pairs, the KLK6/cationic screen provided 13 pairs, and the KLK6/mesotrypsin screen provided 14 pairs (Fig. 3A).

Minor concerns:

Q13

1) The authors provided the Figure 3 which supposed to demonstrate switch effect of all possible pairs of mutations. The picture shows a table where colors present the outcome of mutations. The color is a summed effect of all possible mutations. Why was summation chosen? Why not min or max? It is better to be explained because different mutations at the same position can have a opposite effect, which is shown in Figure 2.

Answer to Q13

The reason for summing the effect of all mutations in Fig. 3A is that we wanted first to identify correlated positions, rather than to identify correlated residues (i.e., specific amino acids). If min or max terms will be used, then the results (i.e., the colors in Fig. 3A) will be dominantly influenced by a specific amino acid pair, which do not necessarily represent the other amino acid pairs within this pair positions. Identifying correlated positions is actually a preliminary stage that allows us to focus on a specific pair for subsequent analysis of correlated residues as shown in Fig. 3B.

The reviewer is right regarding the comment that different mutations at the same position may have an opposite effect. However, summing the ER values in each position allows us to overcome these 'outliers' with opposite effect by identifying positions in which correlated residues are more dominant than non-correlated residues (indicated by the warmer colors in Fig. 3A), and thus these positions are more likely to work cooperatively.

As shown by others, identifying correlated positions in proteins is important to understand protein folding, stability, allostery and catalytic activity and specificity. The hypothesis is that these positions were probably under similar evolutionary constraints (e.g., physical constraints) and thus probably co-evolve (PMID: 27514664, 20862353).

Q14

2) What is fluorescence-activated cell sorting (FACS)? (p.5) Should it have a reference?

Answer to Q14

FACS is a flow cytometer sorting technique that is very widely used for cell sorting; we do not believe a reference is required (in the vast majority of papers, a reference for FACS is not added), but we will add it if the editor thinks it is required. The unabbreviated term was added to the manuscript (page 5).

Q15

3) Figures 2 and 3 have small labels. I would leave the figures as it is but created enlarged copies of them in supplements with better quality.

Answer to Q15

Corrected as suggested. See answer to Q7 (and new Figs. S8 and S9).

Q16

4) Figure S5B does not prove the described statement that T11S/G17R are working in tandem, rather T11S decrease specificity. (p. 12)

Answer to Q16

Indeed, Figure S5B shows that the T11S mutation decreases the selectivity towards KLK6, as compared with the APPI-3M variant [with free energy ($\Delta\Delta G$) of 1.556 kcal/mol], and also in combination with the G17R mutation (G17R in comparison with G17R/T11S with $\Delta\Delta G$ of 0.774 kcal/mol). Yet, although the selectivity of G17R/T11S is decreased compared to G17R, it decreased less than what is expected for T11S alone by -0.782 kcal/mol, which is also the $\Delta\Delta G_{int}$ ($\Delta\Delta G_{int}= 0.774-1.556=-0.782$). The theory says that if the two residues were not working in tandem then the free energy change from APPI-3M to T11S should be equal to the free energy change from G17R to G17R/T11S (i.e., $\Delta\Delta G_{int}=0$) since in both cases we performed the same substitution (i.e., T11S). In our case, $\Delta\Delta G_{int}=-0.782$ and not zero, therefore, we can say that in our case both mutations work in concert, such that T11S is less harmful to selectivity than would have been expected if the two mutations were to work additively (see Table 4, '*Calculated KLK6 total selectivity*'=115 and '*Expected KLK6 total selectivity*'=32; where '*Expected KLK6 total selectivity*' is the expected total selectivity if the two mutations were to work additively and '*Calculated KLK6 total selectivity*' is calculated from the measured values).

Q17

In general, while the study and introduced technique have huge potential, the report looks raw. The authors gave abrupt quantitative analysis and did not provide a broad comparison between single vs. double mutations effect.

The authors concentrated their attention describing an only specific example, leaving the general analysis of mutation landscape without consideration. The analytics in the study should be entirely reorganized, or it should be explained why the only one example is worth to be published.

Answer to Q17

During this study we accumulate a vast amount of data, therefore and as the reviewer noticed, the introduced technique has huge potential that can be interpreted in many types of analyses. Taking into account space and scope limitations, in this study we decided to focus mainly on identifying hotspots, cold spots, specificity-switches, and correlated positions, as these represent features sought after in a wide variety of protein engineering efforts, and we expect that concepts elucidated here will generalize across many different protein systems. We give further rationale for choosing a correlated pair of residues for in-depth analysis in AQ12, above. We decided to give less emphasis to features specific to our specific protein system (although these may be interesting for a protease specialized audience). This study is meant to offer a proof of concept of the methodology (which we have further expanded in the revised manuscript as described in answers to Q1 and Q2 above).

REVIEWERS' COMMENTS:

Reviewer #1 (Remarks to the Author):

The authors have carefully considered the comments, and responded thoroughly. The manuscript is at this point much improved.

In Table 3, the scientific notation makes it very difficult to see trends in the data. It would be better to simply put everything in units of pM, so that the numbers will range from 0.9 to 1060. The same holds for Table 2.

Reviewer #2 (Remarks to the Author):

I am satisfied that the author has adequately addressed the concerns of the reviewers. It is nice that they added a new experiment showing that the technique can be scaled to multiple proteins. I believe it is now suitable for publication.

Reviewer #3 (Remarks to the Author):

I believe that the authors adequately addressed the reviewers' comments.

REVIEWERS' COMMENTS:

Reviewer #1 (Remarks to the Author):

The authors have carefully considered the comments, and responded thoroughly. The manuscript is at this point much improved.

Comment: In Table 3, the scientific notation makes it very difficult to see trends in the data. It would be better to simply put everything in units of pM, so that the numbers will range from 0.9 to 1060. The same holds for Table 2.

Answer: We agree with the reviewer. We changed the Tables accordingly.

Reviewer #2 (Remarks to the Author):

I am satisfied that the author has adequately addressed the concerns of the reviewers. It is nice that they added a new experiment showing that the technique can be scaled to multiple proteins. I believe it is now suitable for publication.

Reviewer #3 (Remarks to the Author):

I believe that the authors adequately addressed the reviewers' comments.